# Pseudoscientific Health Beliefs and the Perceived Frequency of Causal Relationships

**DOI:** 10.3390/ijerph182111196

**Published:** 2021-10-25

**Authors:** Julie Y. L. Chow, Ben Colagiuri, Benjamin M. Rottman, Micah Goldwater, Evan J. Livesey

**Affiliations:** 1School of Psychology, University of Sydney, Sydney, NSW 2006, Australia; ben.colagiuri@sydney.edu.au (B.C.); micah.goldwater@sydney.edu.au (M.G.); evan.livesey@sydney.edu.au (E.J.L.); 2School of Psychology, University of New South Wales, Sydney, NSW 2052, Australia; 3Department of Psychology, University of Pittsburgh, Pittsburgh, PA 15260, USA; rottman@pitt.edu

**Keywords:** pseudoscientific beliefs, contingency learning, causal belief

## Abstract

Beliefs about cause and effect, including health beliefs, are thought to be related to the frequency of the target outcome (e.g., health recovery) occurring when the putative cause is present and when it is absent (treatment administered vs. no treatment); this is known as *contingency learning*. However, it is unclear whether unvalidated health beliefs, where there is no evidence of cause–effect contingency, are also influenced by the subjective perception of a meaningful contingency between events. In a survey, respondents were asked to judge a range of health beliefs and estimate the probability of the target outcome occurring with and without the putative cause present. Overall, we found evidence that causal beliefs are related to *perceived* cause–effect contingency. Interestingly, beliefs that were not predicted by perceived contingency were meaningfully related to scores on the paranormal belief scale. These findings suggest heterogeneity in pseudoscientific health beliefs and the need to tailor intervention strategies according to underlying causes.

## 1. Introduction

Beliefs about cause and effect–or causal beliefs–play a critical role in the decisions individuals make about their health and wellbeing. An individual might choose to give up smoking or to start exercising more because they believe smoking causes long term damage to their health, while exercising improves it. Similarly, an individual might try acupuncture because they believe it effectively relieves pain or take herbal supplements because they believe such remedies improve immune function. Importantly, not all causal beliefs are accurate. Many people come to hold false or pseudoscientific health beliefs, which can be dangerous to their own and others health (e.g., vaccination causes autism; see [1] for a current review). Understanding the psychological mechanisms underlying pseudoscientific health beliefs is crucial to countering them and reducing the substantial individual and societal damage they cause.

According to cognitive psychologists, contingency learning is proposed to be a key mechanism of causal belief formation [2]. It involves the individual making causal judgments based on experiencing or observing the relationship between two events. In a typical laboratory study, participants are presented with a series of fictitious events and asked to judge the strength of the causal relationship between the variables. For example, participants might observe a series of fictitious patients receiving a new drug or no drug and then whether or not their health improves. Presenting the events serially means that the participant acquires the information incrementally, as they might in the real world. The researchers can then compare the participant’s beliefs about the relationship between the events with the objective relationship. In these tasks, people are generally good at accurately judging causal relationships when there is a genuine objective relationship between the events [3,4], e.g., if the fictitious drug is genuinely associated with improvement relative to no drug. However, people are much worse at these tasks when no genuine relationship exists between the events (e.g., illusory causation, see [5] for a review) often tending to overestimate the causal relationship, pointing to systematic biases in contingency learning and causal belief formation.

Researchers have argued that these biases in contingency learning may underpin pseudoscientific beliefs in the real world, including pseudoscientific health beliefs [6,7]. Pseudoscientific beliefs are false causal beliefs that *appear* to be based on facts and evidence but that are not grounded in the scientific method. The possibility that biases in contingency learning may contribute to pseudoscientific beliefs assumes that the same processes observed in the laboratory also apply to causal beliefs in the real world. It assumes that people base their causal beliefs on the extent to which they *perceive* a contingency between events, i.e., their subjective judgement about whether the presence of a potential cause (e.g., a health behaviour) is associated with a change in the likelihood of a particular outcome (e.g., improvement), regardless of whether there is a veridical relationship. Thus, even when there is a no meaningful contingency between behaviour and health improvement, people who report strong belief in the health association should, in theory, also overestimate the frequency of health improvements when the health behaviour is performed relative to when the health behaviour is absent (i.e., positive perceived contingency). However, laboratory studies are far more constrained than everyday life. In real-world settings, the outcome of interest (e.g., health) is often highly variable, meaning that learning the contingency–even when there is one–is far from trivial–especially since we know that contingency learning in the lab can easily be biased by extraneous variables such as the base-rate of the outcome occurring (i.e., outcome density effect [8]) and the frequency of exposure to the putative cause (i.e., cue density effect [3]).

To our knowledge, the relationship between perceived contingency and beliefs about real-world health associations has never been tested, meaning that it is currently unclear whether pseudoscientific health beliefs are even related to biased perceptions of contingency. As an example, many people believe that *Echinacea* is effective for treating the common cold, but rigorous scientific studies indicate it is no more effective than placebo [9]. If these types of real-world pseudoscientific health beliefs are based on perceived contingency, then people who endorse the effectiveness of *Echinacea* in treating the common cold should have—or at least report—more experience recovering quickly from the common cold when *Echinacea* is consumed than when it is not. Understanding whether subjective contingency estimation influences pseudoscientific health beliefs has important implications for the strategies we adopt to overcome them. If people’s pseudoscientific health beliefs are related to the perceived contingency between events, then a potentially effective strategy for correcting these false beliefs is through scientific education—in particular, using base-rate comparison, that is comparing to the frequency of the outcome without intervention [10], to determine the unique influence of the putative cause (e.g., recovery from the common cold in the absence of *Echinacea* use). If pseudoscientific health beliefs are not based on perceived contingency, then other factors should predict them, such as proximity to others who hold such beliefs or other personal factors, and therefore strategies such as base-rate instruction may not be effective in correcting pseudoscientific thinking.

The motivation to address pseudoscientific health beliefs is pertinent as these beliefs are potentially dangerous and costly. In the extreme, choosing an ineffective complementary and alternative medicine (CAM) at the exclusion of scientifically-validated treatments based on pseudoscientific beliefs has resulted in death [11,12,13]. More broadly, in the United States, out-of-pocket expenditure on dietary supplements in 2012 was estimated to be over USD 30 billion dollars [14], despite extensive research suggesting that most dietary supplements are ineffective [15] and even potentially harmful [16].

The purpose of this study was to determine if the perceived contingency between two events, captured by an individual’s memory of personal experiences with a potential cause and a subsequent outcome, is correlated with people’s judgements about a range of contentious health-related beliefs, including beliefs about the efficacy of various complementary and alternative medicines. This study is correlational in nature and thus, to the extent that we find a relationship between causal beliefs and perceived contingency, we cannot determine whether a causal belief is influenced by perceived contingency or vice versa. Nevertheless, merely establishing that the two are associated would be an important demonstration, lending some credibility to the idea that contingency learning in the laboratory is relevant to real-world beliefs. A similar issue relating to self-report measures is the possibility that participants are biased to provide consistent responses when presented with seemingly related questions; participants may provide contingency estimates that are congruent with their belief ratings but are not in fact grounded in actual experience. If this were true, we would expect the size of the correlation between contingency estimates and belief ratings to be the same for all categories of health-related beliefs. Systematic differences in correlation size, however, would suggest that the results cannot be explained entirely by this consistency bias, or would at the very least suggest that participants think that some beliefs should be correlated with perceived contingencies and others should not.

With these issues in mind, we have carefully designed and worded the questions in this survey to reduce any effect of self-report bias when measuring related constructs. Firstly, we counterbalanced the order of question presentation between participants, such that belief ratings preceded contingency estimation questions or vice versa. Secondly, we framed the contingency estimation questions in terms of the people the participant knows in order to avoid participants reporting estimations based on what they have seen or heard on the news, an example of one-shot learning as opposed to incremental accumulation of evidence, which would represent a different kind of learning to what we are interested in. We also explored potential factors that might moderate the relationship between contingency estimation and causal belief, such as the proximity of the person to someone else who endorses these beliefs, and various personality characteristics.

## 2. Method

### 2.1. Participants

A total of two-hundred and ten adult Australians, recruited via Qualtrics Research Services (henceforth referred to as Qualtrics Panel), completed the survey. Potential participants were not informed of the survey content in the initial email invitation. In order to target participants who may have specific beliefs, we geographically targeted two regions: residential areas around wind farms in Australia, and areas with the lowest child-immunisation rates in the country (see Appendix A for list of geographically-targeted postcodes). Of the 210 respondents who completed the survey, 63.3% were female, a majority (70%) were between the ages of 18–40 at the time of participation, 53.3% reported currently being or have been a guardian to young children, and 54.8% resided in metropolitan areas. Participants were provided monetary compensation for completing the survey. A break-down of participants’ demographics is shown in Appendix A.

### 2.2. Design

The survey was a self-report questionnaire involving nine health-related beliefs listed in Table 1. These beliefs were broadly classified into three categories: (1) beliefs about complementary and alternative medicine, (2) controversial beliefs about modern technology and medicine, and (3) beliefs relating to general lifestyle. Some of the beliefs presented in the study are highly controversial and divisive, typically due to overwhelming refutation by scientists (e.g., vaccination–autism), whereas others are more commonly accepted (e.g., exercise–lifespan) or widely used despite mixed evidence (e.g., acupuncture–pain). The inclusion of a wide range of health beliefs that vary both in level of consensus in the population, and amount of direct experience people may have with the health behaviour or outcome, allows for a better understanding of the similarities and differences in the types of information used to inform causal judgements on different types of real-world health beliefs. This also enabled us to determine if the relationship between perceived contingency and causal belief is prevalent across various types of pseudoscientific health beliefs, or if researchers and policymakers need to consider different tailored approaches to understanding what guides a specific false belief and the appropriate strategies to overcoming them.

For each target topic, participants were asked a series of questions aimed at capturing their personal beliefs and experiences with the putative cause (i.e., health behaviour like CAM use) and outcome, as well as their estimates of the probability of the outcome occurring in the presence and absence of the putative cause (indexing the perceived contingencies between events). Critically, we were interested in investigating the relationship between participants’ reported causal belief in the health belief presented, and their estimates of the probability of the outcome occurring with and without the putative cause based on individuals that the participant knows (i.e., perceived contingency). The causal belief rating was formatted to capture the respondent’s endorsement of a causal relationship between the health behaviour and the outcome. An example of this is: *To what extent do you agree that chiropractic therapy* (i.e., health behaviour) *is an effective treatment for back pain* (i.e., outcome)*?* Respondents provided a rating along a sliding scale ranging from 0 (strongly disagree) to 100 (strongly agree) with a midpoint of 50 (neither agree nor disagree). Perceived contingency between the health behaviour and outcome was measured through a series of questions, where respondents were required to provide a percentage estimate of the outcome among all the people they know who have been exposed to the cause, and a percentage estimate of the outcome among all the people they know who have not had exposure to the cause (see Table 2 for an example). We specifically framed these questions in terms of the people the respondent knows in order to avoid them providing responses based on what they might have heard or read in the news that might represent a different kind of learning. A difference score between those two estimates as a proportion (typically represented as Δ*p*) is used as a metric of perceived contingency.

Respondents were also required to complete a series of questions regarding their proximity to the putative cause or outcome (e.g., *select the closest relationship you have to someone who regularly visits a chiropractor*, see Appendix A), how likely they were to recommend or endorse the cause–outcome relationship to others, and how important the health belief is to them personally. These measures were secondary to our primary research question; results from these measures are presented in Appendix A. Together with the causal belief rating, these questions form the belief subcategory, whereas questions requiring respondents to provide some form of probability estimation is categorised into the contingency estimate subcategory. Participants were randomly assigned to receive either the belief subcategory questions first before the contingency estimate questions, or the other way around. We counterbalanced the order of questions, in particular the causal belief ratings and the contingency estimation questions, to account for the possibility that whichever response was given first might influence responses in the subsequent measure. Within each subcategory however, the order of questions remained the same. A summary example of all the different types of questions presented in the survey are described in Table 2.

### 2.3. Procedure

Potential respondents were invited to participate in the survey through an email invitation sent by Qualtrics Panel. Participants were presented with a short demographic questionnaire at the start of the survey, where non-Australian residents were screened out and redirected to an exit screen.

Participants who were eligible for the study were then randomly allocated to one of two order conditions: belief–first or contingency estimate–first. Participants in the belief–first condition received the belief subcategory questions prior to the percentage estimation questions for each health belief. The order was reversed in the contingency estimate–first condition. At the end of the survey, all participants were given a battery of personality measures that have been previously correlated with beliefs in complementary and alternative medicine or illusory causation: modified version of the revised paranormal belief scale [17,18,19], sample 10-item international personality item pool (IPIP) scale [20] measuring conscientiousness and openness to experience based on Costa and McCrae’s [21] NEO personality inventory (NEO-PI-R), and Levenson’s multidimensional locus of control scale [22,23]. The mini-IPIP has previously been validated and found to be psychometrically acceptable for short measures of the big five factors of personality [24].

### 2.4. Statistical Analyses

We were primarily interested in determining whether people’s causal beliefs were predicted by their estimates of the probability of events (i.e., perceived contingency), over and above any personality characteristics that might influence beliefs in pseudoscience or complementary medicine. This was achieved by first computing a perceived contingency (Δ*p*) score for each topic by taking the difference between the participant’s estimation of the probability of the outcome given the cause, p(O|C) (e.g., *Of those people you know who suffer from back pain and visit a chiropractor, what percentage of them have experienced an improvement in their condition?*) and the probability of the outcome given no cause, p(O|~C) (e.g., *Of those people you know who suffer from back pain but do not visit a chiropractor, what percentage of them have experienced an improvement in their condition?*), mirroring Allan’s original contingency equation [25]. This Δ*p* score provides a means of estimating the participant’s perceived contingency between the cue and the outcome.

Using Δ*p* as a predictor variable, we ran a two-step hierarchical regression with causal belief rating as the dependent variable, and mean-centred scores on the personality measures (paranormal belief scale, conscientiousness and openness to experience, and locus of control) as covariates (step one of hierarchical regression). The change in r^2^, Δ*r*^2^, in the regression allows us to estimate the extent to which perceived contingency predicts causal belief, over and above any influence of personality characteristics on illusory causation and pseudoscientific belief.

The regression was conducted on all topic areas separately. Where a Bayes factor is reported, the BF_10_ value is a likelihood ratio of the alternative hypothesis, where there is a meaningful relationship between perceived contingency and causal belief, relative to a null model which includes the four personality measures.

## 3. Results

Due to a presentation error in the questionnaire, causal ratings for the topic diet–feel better were only presented to 62 of the total number of respondents. As a result of this error, we decided to exclude this target topic from further analyses.

### 3.1. Average Causal Belief Ratings

Of the remaining eight target associations, Shapiro–Wilk test of normality revealed that the distribution of causal ratings for most topic areas were not normally distributed, with the exception of two CAM beliefs: chiropractic therapy is an effective treatment for back pain (*p* = 0.053), and acupuncture is an effective treatment for prolonged pain (*p* = 0.066). The lack of normality in the distribution of causal judgements in a majority of these associations is not necessarily surprising considering the target relationships were selected for their divisive, and in the case of vaccination–autism and wind turbine syndrome, controversial nature. However, with the relatively large sample size in our study, the violation of normality assumption should not pose major problems for our analyses [26].

Figure 1 illustrates the average causal belief ratings for all target associations together with the density of responses along the rating scale for each target relationship. As predicted, we found average causal belief ratings to be strongest (reflecting greater consensus on the causal association) for smoking–breathing problems and exercise–lifespan, and weakest for controversial beliefs such as vaccination–autism, wind turbine syndrome, and WIFI–cancer. Belief ratings for popular complementary treatments and therapies fell in between the two former categories; causal ratings for herbal–cold, chiropractic–backpain and acupuncture–pain were more evenly distributed across the entire scale, with average ratings closer to the midpoint of the scale.

### 3.2. Average Contingency Estimate

We were unable to calculate the perceived contingency for wind turbine syndrome since it was not intuitive to question respondents on the proportion of people they know who do not live close to a wind farm but who have experienced adverse health effects. Unlike the other health beliefs, the target effect for wind turbine syndrome is not a discrete event but a constellation of highly common symptoms; there would be too many alternative causes for common symptoms such as headaches and disturbed sleep. Instead, we asked respondents to estimate of the probability of no outcome (no adverse health effects) given putative cause (wind turbines). However, since these measures are not equivalent, we have omitted them from our analysis).

For the remaining seven target associations, we computed a Δ*p* score as a metric for perceived contingency by taking the difference in estimation for the probability of the outcome in the presence of the putative cause and in its absence (see *Statistical Analyses)*. The calculation of a Δ*p* score was only possible when the respondent has provided ratings for both frequency estimate questions. Participants who reported 100% of the people they know experiencing the outcome with the health behaviour were not presented with the complementary probability question (outcome in the absence of the health behaviour). In the case of vaccination–autism and WIFI–cancer, the scale provided to make a probability estimate was from 0–5% (autism) and 0–10% (cancer) to reflect actual population prevalence. This provided us with Δ*p* scores ranging from −1 (outcome only occurs in the absence of the cause) to + 1 (outcome only occurs in the presence of the cause) on all topics. Negative Δ*p* suggests the outcome is rated as more likely to occur in the absence of the health behaviour than in its presence. An illustration of the distribution of mean Δ*p* estimates for each target topic and the corresponding frequency ratings for the outcome with and without the putative cause present is depicted in Figure 2.

### 3.3. Contingency Estimation and Causal Belief

The primary purpose of this study was to investigate the relationship between participants’ perceived contingency between the putative cause and the outcome, and their causal beliefs. Using a hierarchical regression with causal belief ratings as the dependent variable and Δ*p* score as the predictor variable (controlling for scores on various personality measures), we found a significant relationship between Δ*p* and causal belief, with regressions from five of the seven topics showing contingency estimates to be a significant predictor of causal belief ratings: herbal–cold (Δ*r*^2^ = 0.043, Δ*F* (1180) = 8.53, *β* = 0.213, *p* = 0.004, BF_10_ = 13.3), chiropractic–backpain (Δ*r*^2^ = 0.264, Δ*F* (1149) = 59.4, *β* = 0.528, *p* < 0.001, BF_10_ = 2.54e + 9), acupuncture–pain (Δ*r*^2^ = 0.106, Δ*F* (1117) = 15.8, *β* = 0.345, *p* < 0.001, BF_10_ = 235), smoking–breathing problems (Δ*r*^2^ = 0.103, Δ*F* (1193) = 23.1, *β* = 0.079, *p* < 0.001, BF_10_ = 5721), and exercise–lifespan (Δ*r*^2^ = 0.179, Δ*F* (1158) = 37.8, *β* = 0.436, *p* < 0.001, BF_10_ = 1.49e + 6). These relationships were in the expected direction, indexed by positive beta values: increase in Δ*p* estimates predicted stronger causal belief ratings (i.e., stronger agreement that the health behaviour causes the outcome). These findings, illustrated in Figure 3, provide novel preliminary evidence for the relationship between contingency learning and causal beliefs outside the laboratory with real-world health beliefs, suggesting that at least for two of the three classes of beliefs that we investigated, people’s causal beliefs were related to their perceptions of the frequencies of events that their acquaintances have experienced. To determine if the order of presentation (contingency subcategory first or belief subcategory first) influenced the relationship between causal beliefs and Δ*p*, we also conducted the analysis separately for participants who saw the causal belief ratings first, and those who saw the contingency estimation questions first. These results are reported in Appendix A. Overall, there was no systematic difference in the order of question presentation on the results, and therefore, the results reported here collapse across question order.

There was no significant relationship between contingency estimates and causal beliefs for the relationship between vaccination and autism (Δ*r*^2^ = 0.008, Δ*F* (1133) = 1.46, *β* = 0.094, *p* = 0.229, BF_10_ = 0.545), and radiation from mobile phones and Wi-Fi causing cancer (Δ*r*^2^ < 0.001, Δ*F* (1165) = 0.014, *β* = 0.005, *p* = 0.906, BF_10_ = 0.364). As a secondary analysis, we determined whether the relationship between perceived contingency and causal belief was moderated by the proximity of the respondent to someone else who engages in the health behaviour (e.g., visits a chiropractor). Proximity questions were asked in terms of the *cause* for all health beliefs other than vaccination–autism, where the question was framed in terms of proximity to the *outcome* (i.e., someone with autism). This was decided because proximity to someone with autism was thought to be more informative than proximity to someone who has been vaccinated. Here, we found no consistent evidence that the relationship was influenced by proximity (see Appendix A). On the whole, our findings suggest that people’s causal beliefs were related to their estimations of the probability of cause–outcome events.

### 3.4. Individual Differences Measures

All participants completed three questionnaires at the end of the survey measuring different personality factors that may influence their causal beliefs: the revised paranormal belief scale (rPBS; [18]), measures of conscientiousness and openness to experience [20] and Levenson’s multidimensional locus of control scale [22,23]. Mean scores on each of the ID measures and how each score was calculated are presented in Appendix A. For the full questionnaire of each personality measure, see Appendix A.

We were interested in determining the relationship between respondents’ scores on each of the personality questionnaires separately on causal belief ratings for each target health belief, controlling for scores on all other personality measures. These results are presented in Table 3. Bonferroni corrected *p*-value = 0.0125. Overall, we found that participants’ score on the rPBS was a predictor of causal ratings for controversial beliefs relating to modern technology and medicine, in particular their beliefs about vaccination–autism and WIFI–cancer.

We found participants’ scores on the openness to experience scale to be a significant predictor for the causal ratings for herbal–cold, acupuncture–pain, and exercise–lifespan. All other analyses on this measure were not statistically significant. On measures of conscientiousness and Levenson’s multidimensional locus of control, we did not find evidence that scores on these questionnaires significantly predict causal ratings in any of the health beliefs, all *p* > 0.0125.

## 4. Discussion

In this survey, we were primarily interested in whether people’s belief in a range of (pseudo)medical health associations were meaningfully related to their perceptions of the contingency between the health behaviour and the occurrence of the target outcome. Overall, we found strong evidence that contingency estimates correlate meaningfully with causal judgements, especially for beliefs about complementary and alternative medicine. Additional analyses on the personality measures, in particular scores on the paranormal belief scale, also suggest a role of personality factors in influencing causal beliefs for contentious associations about the negative effects of modern technology and medicine (WIFI and vaccination). This category of beliefs concerns putative causes and effects that have extreme frequency of exposure, whether it be very high frequency (WIFI, vaccination), or very low frequency (wind turbines, autism, cancer). This means that most people are unlikely to gain quality contingency information from their own experiences or the experiences of close friends and relatives. It is thus noteworthy that causal beliefs for these relationships, in particular for vaccination–autism and WIFI–cancer, were well predicted by belief in the paranormal but also poorly predicted by contingency estimation.

The primary research question in this study was whether causal ratings were related to contingency estimates based on perceived frequencies of events among all the people the participant knows. Our findings suggest that perceived contingency was predictive of their causal belief on most measures, and this relationship was most reliable for health beliefs relating to CAM (herbal remedies, chiropractic therapy, acupuncture), and beliefs about healthy lifestyle that are widely endorsed (smoking causes breathing problems, exercise increases lifespan). To our knowledge, this is the first demonstration of the consistency between people’s contingency estimates and causal judgements on real-world health beliefs. Although we cannot ascertain causal direction from this correlational study, the results could mean that people acquire beliefs about cause–effect relationships by observing relationships between health behaviours and their supposed effects. This finding parallels contingency learning research in the laboratory that uses CAM-like cover stories, where participants use information obtained through trial-by-trial learning about the likelihood of recovery given a novel treatment and no treatment to judge the efficacy of the treatment in treating the disease (e.g., [8] and [27]).

One important consideration when interpreting these findings is the reciprocal nature of the relationship between contingency estimates and causal beliefs, at least in the way they are measured in this study, and in fact when it comes to assessing these relationships in real-world beliefs. We theorize based on experimental research that the perceived probability of events is causal in influencing the beliefs that people hold. However, it is also possible for the causal relationship to be in the opposite direction, where people use their existing beliefs to retrospectively estimate the probability of events to align with their beliefs. To control for the possibility that people may be calibrating their estimates of event frequencies to be internally consistent with their causal beliefs, we counterbalanced the order of the two sets of questions between respondents. Appendix A revealed no convincing evidence that the order in which the questions were presented affected the strength of the correlation between causal belief ratings and contingency estimates. That is, there is no evidence that people who were asked to provide a causal judgement first were more likely to inflate their probability estimates to be more consistent with their reported beliefs. In any case, the argument that participants provided estimates congruent with their causal beliefs in order to maintain some internal consistency does not explain why the contingency–belief relationship was present for some health beliefs but not others. Nevertheless, this study provides evidence that at least for some causal beliefs, including beliefs about CAM that are commonly used despite mixed evidence, perceived contingency between treatment use and health outcomes are meaningfully related to people’s judgements of treatment efficacy. An important question for future research is whether promoting strategies that improve the estimation of the probability of events also improve people’s ability to infer a null causal relationship when there is none.

One explanation for why people might perceive a positive contingency where none exists is the tendency for people to overweigh instances where the putative cause and the target outcome are both present (e.g., taking *Echinacea* and recovering from the cold), than when either the putative cause or the target outcome is absent (or both are absent). Experimental studies have shown that manipulations that increase cause–outcome coincidences are particularly effective at producing stronger judgements about the causal relationship between the two events, regardless of whether the two events are actually causally associated with each other [28,29]. Similarly, there is considerable evidence that under certain conditions people do not use base-rate information appropriately during causal induction [30]. These biases encourage an overestimation of the contingency between the putative cause and the target outcome, resulting in the development of strong false causal beliefs.

It is also worth mentioning here that even though laboratory research has consistently shown that causal beliefs are largely derived from covariational information, the reverse is also possible—that is, existing causal beliefs influence how information is processed. One potential way in which causal beliefs might influence contingency estimation, is through a process such as confirmation bias. Confirmation bias is the tendency for people to selectively recall information that is consistent with their existing beliefs [31]. For instance, someone entertaining the belief that herbal remedies are effective for treating the common cold is more likely to recall occasions where the remedy was taken and the individual recovered, than instances where they recovered without any intervention. This is all to say that although it is possible that some people experience a positive contingency by chance where an objective causal relationship is non-existent, it is not necessary to assume that this is the case since causal illusions are anticipated to arise even under zero-contingency scenarios, or through a process such as confirmation bias.

The exceptions to the contingency–belief relationship are noteworthy; there is no convincing evidence of a relationship between contingency estimates and causal ratings for the belief that childhood vaccination causes autism, and exposure to radiation from mobile phones and WIFI causes cancer. In these cases, people’s causal beliefs may not be related to the information gained from observing the co-occurrence of events in the environment, implying that their causal beliefs are driven by other sources of information. An important characteristic of these health beliefs is that both involve a putative cause that is so prevalent that there is inadequate sampling of cause–absent events (low frequency of people who have not been vaccinated, and low frequency of people who are not exposed to mobile phones and WIFI), as well as a low overall probability of the outcome (autism, cancer). Perhaps then when the quality of first-hand information is particularly poor, and when there is a considerable amount of secondary information in various media, causal beliefs are based much more strongly on factors other than the perceived contingency between events. For example, infrequent occurrences of the outcome would make it especially difficult to amass evidence from everyday life against the putative relationship that the individual has read or heard on the news (an example of one-shot learning). This might have implications for the way that we go about tackling these beliefs relative to the others that are consistent with contingency estimation.

Current efforts to overcome misinformation typically rely on providing an alternative causal explanation for the outcome (see [32] for a review); researchers are challenged to consider the generalizability of these strategies across different types of false beliefs. A study evaluating the effectiveness of corrective information on vaccination beliefs found that the strategy was ineffective in improving misperceptions about vaccine safety among respondents with high levels of concern about the side effects of vaccination, and further reduced their intention to vaccinate [33]. Results from our study suggest that strategies that highlight the statistical relationship between events as a means of inferring causal relationship may not be useful in changing *some* pre-existing beliefs, but useful for others. For example, beliefs about the effectiveness of herbal supplements are related to misinterpretations or misperceptions of contingencies, and so providing ways to re-interpret the contingencies should be more helpful in these cases (e.g., [34]). However, when individual experiences are insufficient for accurate contingency estimation, due to extremely high or low frequency of exposure to the putative cause or outcome, individuals are better off deferring to the findings of scientific studies or the opinions of professionals. Future research should investigate how people come to form strong beliefs about things they have little experience with, and how beliefs formed in this manner may be different to those formed through extensive direct experience. Altogether, our data suggest the heterogeneity in the causes that underpin pseudoscientific beliefs require us to consider more comprehensive, and tailored strategies that account for these differences. In order to achieve this, there is an imperative to first identify which pseudoscientific beliefs are a product of perceived contingency, and which are due to other factors such as personality characteristics.

Results from the four personality measures showed that participants who scored highly on the openness to experience measure were more likely to endorse health beliefs related to improving health outcomes, such as CAM for the relief of the common cold and prolonged pain, and exercise to increase lifespan. This finding may not be surprising, since people who are open to new experiences may be more attracted to a wide range of novel approaches to improving health. More interestingly, we found participants’ scores on the revised paranormal belief scale (rPBS) predict causal ratings on a subset of beliefs relating to negative effects of modern technology, such as vaccination causes autism and WIFI causes cancer. These were also beliefs that were not well predicted by perceived contingency between the putative cause and the outcome. In these cases, there is reason to assume that participants’ contingency estimates are unreliable; the probability of someone they know being exposed to WIFI and have been vaccinated are both incredibly high. In these cases, then, it seems personality factors may be a stronger determinant of causal beliefs. Within the contingency learning literature, paranormal belief has been linked to heightened illusory causation, where participants who report strong beliefs in the paranormal are more likely to expose themselves to cause–present information [35], as well as lower signal-detection criterion—strong believers in the paranormal were more likely to draw associations between unrelated stimuli [36] or perceived there to be a causal agent for events where there is none [37]. Taken together, it is possible that when there is insufficient direct experience with the putative cause and outcome for accurate contingency estimation, participants who are susceptible to biased thinking are more likely to expose themselves to confirmatory evidence and/or interpret ambiguous events as evidence for the causal relationship, thus they develop more positive opinions about these beliefs. Additionally, it may also be the case that in the absence of any direct experience, people who believe in the paranormal are also more likely to accept and endorse spurious beliefs that suggest a causal association between events. Controversial beliefs such as vaccination causes autism and WIFI exposure causes cancer may also be considered conspiratorial beliefs. It is possible that people who endorse these beliefs also believe in shadow entities (e.g., ‘Big Pharma’ and ‘Big Tech’), who are responsible for concealing confirmatory evidence for personal gain, therefore, “true” contingencies are not readily detectable by the average person. Although outside the scope of the current study, future research could investigate the relationship between endorsement of conspiratorial beliefs and the dissociation between contingency learning and causal belief.

## 5. Conclusions

In summary, we found a meaningful relationship between the perceived frequency of the outcome occurring in the presence and absence of the putative cause (i.e., contingency learning), and judgements of causality across a range of health beliefs, including popular complementary and alternative medicine and therapies. To our knowledge, this is the first demonstration of the relationship between contingency estimation and causal judgement on real-world health beliefs, in particular beliefs relating to CAM and judgements of treatment efficacy. This finding is promising as it suggests that strategies that effectively improve people’s ability to accurately infer the likelihood of recovery from an illness with and without the alternative therapy should thus change their beliefs about the efficacy of the treatment when used for that purpose. However, when there is inadequate sampling of information, due to exceedingly low exposure to cause–absent events and/or low overall probability of the outcome occurring, causal beliefs were not meaningfully correlated with contingency estimates. In these cases, we also found that the tendency to believe in the paranormal was a strong predictor of causal judgements, suggesting that without sufficient first-hand information, personality characteristics may play a strong role in determining whether people are likely to detect a causal relationship where none exists.

The findings in this survey may be important for understanding the types of information that are used to inform judgements about a range of health beliefs. Critically it highlights differences between beliefs; beliefs in CAM are meaningfully related to the perceived contingency between treatment use and recovery, however controversial beliefs about modern medicine and technology appear to be better predicted by personality factors than perceived event frequencies. These differences may influence the kinds of strategies that are effective in challenging false causal beliefs.

## Figures and Tables

**Figure 1 ijerph-18-11196-f001:**
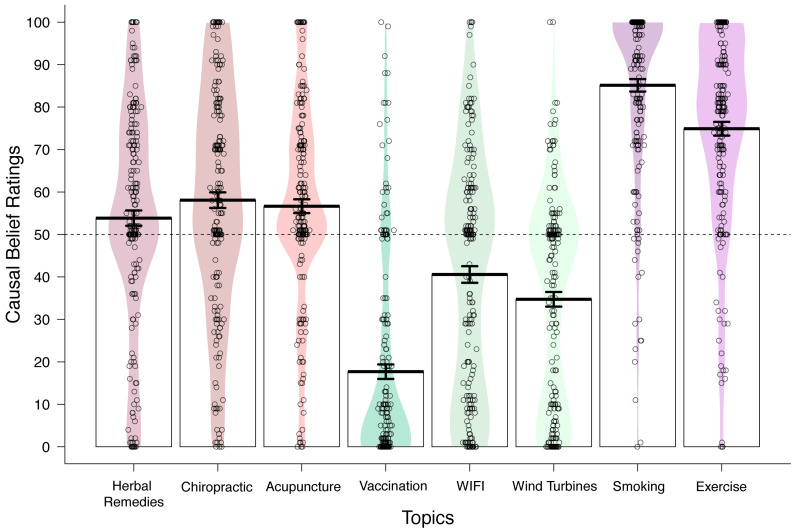
Plot of the density of responses (*n* = 210) on the causal belief question for each target topic area grouped by belief type. Each circle represents a participant’s response, and the coloured region reflects the density of the responses along the *Y*-axis. Mean causal rating (±SE) across all respondents is denoted by a bold black line.

**Figure 2 ijerph-18-11196-f002:**
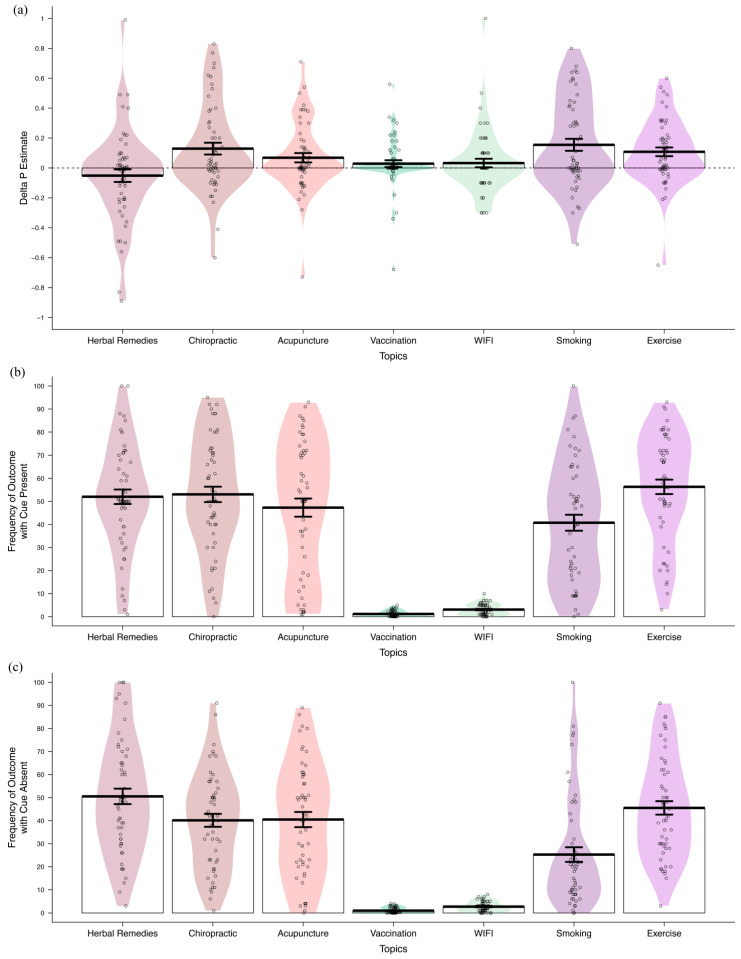
(**a**) Delta *p* estimates for each target topic, measured as a difference in the probability of the outcome occurring in the presence and absence of the target health behaviour. (**b**) Reported frequency (from 0–100%) of the target outcome occurring in the presence of the cause, and (**c**) in the absence of the cause. Frequency estimates for autism were made from a scale of 0–5%, and cancer estimates were made from a scale of 0–10%. Coloured region denotes the density of the estimates along the *Y*-axis. Labels on the *X*-axis denotes the topic area by cue name. Delta *p* estimates were only calculated for participants who provided both outcome frequency estimates (with and without the cause present).

**Figure 3 ijerph-18-11196-f003:**
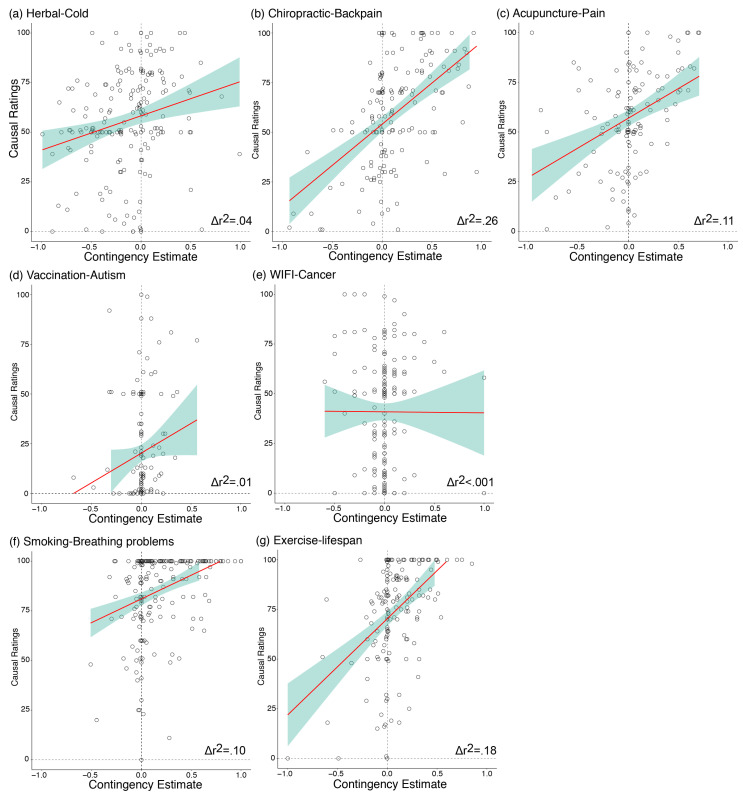
Scatterplot with regression line for each individual participant’s causal belief rating (standard error shaded in green) as a function of the calculated contingency estimate (Δ*p*) across all topic areas. A significant relationship between the two factors was found for all topics with the exception of vaccination–autism (Figure 3d) and WIFI–cancer (Figure 3e).

**Table 1 ijerph-18-11196-t001:** Target health beliefs included in this study.

Topic Area
**Complementary and Alternative Medicine**
Herbal remedies are effective in treating the common cold (herbal–cold)
Chiropractic therapy is an effective treatment for backpain (chiropractic–backpain)
Acupuncture is an effective treatment for prolonged pain (acupuncture–pain)
**Controversial beliefs about modern technology and medicine**
Childhood vaccination causes autism (vaccination–autism)
Exposure to wind turbines can cause adverse health effects including memory loss, disturbed sleep and headaches (even when the turbine cannot be heard) (wind turbine syndrome)
Radiation from mobile phones and WIFI causes cancer (WIFI–cancer)
**Beliefs relating to general lifestyle**
Prolonged smoking causes heart, lung and breathing problems (smoking–breathing)
Regular exercise increases lifespan (exercise–lifespan)
Practicing a restrictive diet can make you feel better (diet–feel better)

Note: Shorthand notation for each causal belief is presented in parentheses and will be used to denote the respective causal association throughout the paper.

**Table 2 ijerph-18-11196-t002:** Target questions presented in each sub-category with chiropractic–backpain as an example topic. A full list of questions presented in each sub-category can be found in Appendix A.

Belief Sub-Category
Belief *To what extent do you agree that chiropractic therapy is an effective treatment for back pain?*
**Contingency Estimate sub-category**
Probability of the cause, *c* *Of all the people you know, what percentage of people visit a chiropractor for back pain?*
Estimate of probability of outcome given cause, p(O|C) *Of those people you know who suffer from back pain and visit a chiropractor, what percentage of them have experienced an improvement in their condition?*
Estimate of probability of outcome given no cause, p(O|~C) *Of those people you know who suffer from back pain but do not visit a chiropractor, what percentage of them have experienced an improvement in their condition?*

**Table 3 ijerph-18-11196-t003:** Summary statistics of the relationship between the four personality variables measured at the end of the survey (revised paranormal belief scale, measures of openness to experience and conscientiousness, and Levenson’s multidimensional locus of control scale) on causal ratings.

	Revised Paranormal Belief Scale	Conscientiousness	Openness to Experience	Locus of Control
Herbal–Cold	β = 0.082, *t*(4) = 1.12, *p* = 0.263 BF_10_ = 0.497	β = 0.149, *t*(4) = 2.18, *p* = 0.030 BF_10_ = 2.42	β = 0.181, *t*(4) = 2.65, *p* = 0.008 * BF_10_ = 6.81	β = 0.046, *t*(4) = 0.64, *p* = 0.523 BF_10_ = 0.335
Chiropractic–Backpain	β = 0.148, *t*(4) = 1.98, *p* = 0.049 BF_10_ = 1.76	β = 0.020, *t*(4) = 0.28, *p* = 0.777 BF_10_ = 0.307	β = 0.128, *t*(4) = 1.85, *p* = 0.066 BF_10_ = 0.351	β = 0.030, *t*(4) = 0.41, *p* = 0.681 BF_10_ = 0.320
Acupuncture–Pain	β = 0.216, *t*(4) = 2.99, *p* = 0.003 * BF_10_ = 15.2	β = 0.065, *t*(4) = 0.97, *p* = 0.333 BF_10_ = 0.402	β = 0.246, *t*(4) = 3.67, *p* < 0.001 * BF_10_ = 108.9	β = −0.014, *t*(4) = −0.20, *p* = 0.845 BF_10_ = 0.264
Vaccination–Autism	β = 0.338, *t*(4) = 4.82, *p* < 0.001 * BF_10_ = 6.83e + 3	β = 0.033, *t*(4) = 0.50, *p* = 0.620 BF_10_ = 0.829	β = −0.107, *t*(4) = −1.64, *p* = 0.102 BF_10_ = 0.591	β = 0.061, *t*(4) = 0.88, *p* = 0.380 BF_10_ = 0.338
Wind Turbine Syndrome	β = 0.147, *t*(4) = 2.01, *p* = 0.046 BF_10_ = 1.76	β = 0.073, *t*(4) = 1.06, *p* = 0.292 BF_10_ = 0.467	β = −0.080, *t*(4) = −1.17, *p* = 0.242 BF_10_ = 0.190	β = 0.137, *t*(4) = 1.88, *p* = 0.062 BF_10_ = 1.41
WIFI–Cancer	β = 0.254, *t*(4) = 3.46, *p* < 0.001 * BF_10_ = 56.9	β = 0.074, *t*(4) = 1.07, *p* = 0.286 BF_10_ = 0.477	β = 0.062, *t*(4) = 0.91, *p* = 0.364 BF_10_ = 0.110	β = 0.044, *t*(4) = −0.61, *p* = 0.545 BF_10_ = 0.334
Smoking–Breathing Problems	β = −0.087, *t*(4) = −1.17, *p* = 0.245 BF_10_ = 0.558	β = 0.069, *t*(4) = 0.98, *p* = 0.326 BF_10_ = 0.467	β = 0.117, *t*(4) = 1.68, *p* = 0.095 BF_10_ = 0.565	β = −0.018, *t*(4) = −0.24, *p* = 0.808 BF_10_ = 0.309
Exercise–Lifespan	β = 0.047, *t*(4) = 0.65, *p* = 0.517 BF_10_ = 0.331	β = 0.116, *t*(4) = 1.70, *p* = 0.091 BF_10_ = 1.02	β = 0.234, *t*(4) = 3.44, *p* < 0.001 * BF_10_ = 54.5	β = −0.076, *t*(4) = −1.06, *p* = 0.293 BF_10_ = 0.456

* *p* < 0.0125 and BF_10_ > 3; Bayes Factors are compared to a null model which includes scores on the other three personality measures.

## Data Availability

The datasets generated and/or analysed during the current study are available in the Open Science Framework repository, https://osf.io/69yeg/ (accessed on 28 July 2020). The study was not pre-registered.

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
