# Peer review of "Pseudoscientific Health Beliefs and the Perceived Frequency of Causal Relationships"

_ijerph, 2021, doi:10.3390/ijerph182111196_

Round 1

Reviewer 1 Report

The paper reports a single survey study of the relationship between health beliefs and perceived contingency between the putative cause and the health outcome.  I found the paper interesting.  Although it is not directly in my area, I found it reasonably easy to follow.  It had a strong justification in drawing on lab-based research on contingency learning and it was well-conducted, and the analyses were appropriate and very thorough.  The results were also interesting but highlight that these beliefs do not have a simple causation.  This was acknowledged in the introduction and built into the methodology by asking participants to think only about people they knew, so it was perhaps not surprising when the relationships found were relatively small.  Nevertheless, the pattern was interesting, and particularly with the separation of the pattern of relationship for paranormal belief with that of the contingency data.  I do not have any revisions to suggest.

Author Response

We thank Reviewer 1 for their kind feedback.

Reviewer 2 Report

In this article the authors address a very complicated, but very important, research question. They investigated if it is true that pseudoscientific beliefs are somehow produced by biases in human contingency learning. This idea has been proposed numerous times, but never directly tested. The authors put this hypothesis to the test in this very ingenious study.

This study is a first approach to the problem, and obviously has limitations. However, its main limitations are identified by the authors and properly addressed/commented in the article (e.g. correlation does not mean causation). The resulting article is excellent, and I think it makes a very needed contribution to the field.

However, I have some concerns. None of them is critical, but I honestly think that by addressing them the authors may improve the article.

  • The authors said “[…] we framed the contingency estimation questions in terms of the people the participant knows in order to avoid participants reporting estimations based on what they have seen or heard on the news, an example of one-shot learning as opposed to incremental accumulation of evidence, which would represent a different kind of learning to what we are interested in.” This makes sense because they were relating contingency learning biases with pseudoscientific beliefs. But I cannot help but thinking that a lot of what we belief is probably influenced by this type of one-shot learning. This is probably more important for those topics in which having in-person experience is really infrequent. Please consider adding some discussion about this idea (probably in the Discussion).
  • In my opinion, it is not sufficiently clear why the authors were unable to calculate the perceived contingency for ‘wind turbine syndrome’. They said in a footnote “there would be too many alternative causes for common symptoms like headaches and disturbed sleep. Instead we asked respondents to estimate of the probability of no outcome (no adverse health effects) given putative cause (wind turbines). However since these measures are not equivalent, we have omitted them from our analysis” I do not follow, sorry. As I see it, there other beliefs in which the ‘effects’ have many alternative causes (e.g., cancer may be produced by many causes). Thus, I do not understand why the authors made an exception with ‘wind turbine syndrome’.
  • The non-significant regressions (vaccination-autism and WIFI-cancer) yielded Bayes Factors indicating no good evidence for the alternative nor the null hypothesis. Considering that, the following idea is not completely true “there appears to be next to no relationship between contingency estimates and causal ratings for the belief that childhood vaccination causes autism, and exposure to radiation from mobile phones and WIFI causes cancer.”
  • I have some concerns about how FWER was corrected. The authors used Bonferroni for their analysis for the questionnaires (e.g. Table 3). However, they did not control FWER in their regressions (although there were seven of them). Also, I wonder why they decided to control FWER by using Bonferroni, when there are other solutions which are more appropriate (e.g., Holm-Bonferroni). Finally, the analysis of the questionaries involved 32 correlations. The p-value for considering a result significant was set at .0125. However, if you applied Bonferroni, that threshold should be .00156.
  • From the Discussion: “One explanation for why people might perceive a positive contingency where none exists is the tendency for people to overweigh instances where the putative cause and the target outcome are both present” In order to investigate this interesting idea, the authors can make separate regressions for their p(O|C) and p(O|~C) data. The rationale for doing that is to find out if one of these terms alone can explain part of the variance from the causal belief data.
